# The Impact of Chronic Kidney Disease on the Mortality Rates of Patients with Urological Cancers—An Analysis of a Uro-Oncology Database from Eastern Europe

**DOI:** 10.3390/jpm13111572

**Published:** 2023-11-03

**Authors:** Mircea Ciorcan, Șerban Negru, Răzvan Bardan, Alin Cumpănaș, Iasmina Mattar, Yahya Bitar, Lazăr Chișavu, Luciana Marc, Adalbert Schiller, Adelina Mihăescu

**Affiliations:** 1Department of Clinical Practical Skills, Victor Babes University of Medicine and Pharmacy, 300041 Timișoara, Romania; ciorcan.mircea@umft.ro; 2Center of Advanced Research in Cardiovascular Pathology and Hemostaseology, Victor Babes University of Medicine and Pharmacy, 300041 Timișoara, Romania; 3Department of Oncology, Victor Babes University of Medicine and Pharmacy, 300041 Timișoara, Romania; serban.negru@umft.ro; 4Oncohelp Oncology Center, 300239 Timișoara, Romania; 5Department of Urology, Victor Babes University of Medicine and Pharmacy, 300041 Timișoara, Romania; cumpanas.alin@umft.ro; 6Department of Urology, Clinical Emergency County Hospital, 300723 Timișoara, Romania; dr.iasmina.mattar@gmail.com (I.M.); bitar.yahya.almansour@gmail.com (Y.B.); 7Department of Nephrology, Victor Babes University of Medicine and Pharmacy, 300041 Timișoara, Romania; chisavu.lazaru@umft.ro (L.C.); marc.luciana@umft.ro (L.M.); schiller.adalbert@umft.ro (A.S.); mihaescu.adelina@umft.ro (A.M.); 8Department of Nephrology, Clinical Emergency County Hospital, 300723 Timișoara, Romania; 9Center for Molecular Research in Nephrology and Vascular Disease, Victor Babes University of Medicine and Pharmacy, 300041 Timișoara, Romania

**Keywords:** chronic kidney disease, overall mortality, prostate cancer, bladder cancer, renal cell carcinoma

## Abstract

(1) Background: The relationship between chronic kidney disease (CKD) and urological cancers is complex, as most of these cancers are diagnosed in patients with advanced ages, when the kidney function may be already impaired. On the other hand, urological cancers could represent a risk factor for CKD, significantly reducing the life expectancy of the patients. The main objective of our study was to analyze the impact of CKD on the overall mortality of patients diagnosed with the most frequent types of urological cancers. (2) Material and Methods: We conducted an observational retrospective cohort study on a group of 5831 consecutive newly diagnosed cancer patients, followed over a 2-year period (2019–2020), from a large Oncology Hospital in Romania. From this group, we selected only the patients diagnosed with urological malignancies, focusing on prostate cancer, bladder cancer and renal cancer; finally, 249 patients were included in our analysis. (3) Results: In the group of patients with prostate cancer (*n* = 146), the 2-year overall mortality was 62.5% for patients with CKD, compared with 39.3% for those with no initial CKD (*p* < 0.05). In the group of patients with bladder cancer (*n* = 62), the 2-year overall mortality was 80% for patients with initial CKD, compared with 45.2% for the patients with no initial CKD (*p* < 0.05). Finally, in the group of patients with renal cell carcinoma (*n* = 41), the 2-year overall mortality was 60% for patients with initial CKD, compared with 50% for the patient group with no initial CKD (*p* < 0.05). Various correlations between specific oncologic and nephrological parameters were also analyzed. (4) Conclusions: The presence of CKD at the moment of the urological cancer diagnosis is associated with significantly higher 2-year mortality rates.

## 1. Introduction

The relationship between chronic kidney disease (CKD) and urological cancers is complex. On the one hand, most of the cancers are diagnosed in patients with advanced ages, when the kidney function may be already impaired due to hypertension, cardiovascular disease or diabetes mellitus, or because of the normal aging process [1,2]. On the other hand, urological cancers could represent a risk factor for CKD, due to various nephrotoxic interventions or paraneoplastic syndromes, or due to the presence of cancer per se (especially in renal cancer) [3,4].

Prostate cancer has an impact on kidney function in some situations: as the prostate grows due to locally advanced cancer, it can cause bladder outlet obstruction or invade the ureteric orifices [5,6]. On the other hand, patients with CKD diagnosed with prostate cancer may have reduced kidney function, and some chemotherapy drugs (such as docetaxel and cabazitaxel, indicated in metastatic disease) may need to be adjusted or even avoided [4,7].

Bladder cancer can also have a negative impact on kidney function: large tumors may invade the ureter orifices and cause hydronephrosis and subsequent kidney damage [8]. Similarly, renal cell carcinoma may represent a cause for chronic kidney disease: initially, tumor growth inside the kidney can compress the surrounding kidney tissue, reducing the functional filtration capacity. Subsequent venous thrombus development could cause significant vascular obstruction, with blood flow impairment and kidney ischemia, leading to irreversible damage [9]. Moreover, radical nephrectomy, which is still widely performed, increases the incidence of CKD, especially if the remaining kidney already had a degree of damage. CKD could also be associated with a higher risk of surgical complications, related to anesthesia, wound healing and postoperative recovery [10]. All these factors have an impact on the overall prognosis and mortality rates.

The aim of our study was to analyze the impact of CKD on the overall mortality of patients diagnosed with the most frequent types of urological cancers. We also analyzed some potential correlations between specific clinical parameters and the outcomes of the patients.

## 2. Materials and Methods

We selected our patients from a total number of 5831 consecutive newly diagnosed cancer patients, over a 2-year period (1 January 2019–31 December 2020), from a large Oncology Hospital and Outpatient Department in western Romania, as previously reported [11].

From the mentioned database, we selected the patients with urological malignancies initially diagnosed in the Department of Urology of Timisoara Clinical Emergency County Hospital (the most significant Urology Department in the area), focusing on cases of prostate cancer, bladder cancer and renal cancer. All patients underwent a form of cancer therapy before or after their inclusion in the analysis, including ablative surgery, radiotherapy, hormone therapy, chemotherapy or immunotherapy. Patients with testicular cancer, urothelial cancer of the upper urinary tract, urethral cancer and penile cancer were excluded from the analysis, as they represented only a small proportion of the total sample of patients with urological malignancies from the mentioned Urology Department, reducing the statistical significance of the collected data.

The study was approved by the Ethics Committee of the Oncohelp Oncology Center (Approval Number 2/9 July 2020). All included patients signed an informed consent form, in which they gave their approval for the use of their clinical data for scientific purposes, under the condition of strict confidentiality. Consequently, the identity of the patients was blinded to the investigators, using a computer-assigned unique identification number.

Before the presentation in the Department of Urology for suspicion of a urological malignancy, no patient was known to have chronic kidney disease (CKD). Patients with previously diagnosed CKD (and eventually on hemodialysis) were not included in our analysis. Based on the baseline assessment of the renal function performed at the time of the initial oncological diagnosis, we divided the patients into two groups: the first group included patients with CKD, and the second (control) group included the patients with no CKD. CKD was defined with an Estimated Glomerular Filtration Rate (eGFR) < 60 mL/min/1.73 m^2^, persistent for more than 90 days, according to the 2012 Kidney Disease—Improving Global Outcomes (KDIGO) Guidelines [12]. The eGFR levels were calculated using the Chronic Kidney Disease Epidemiology Collaboration (CKD-EPI) formula. The first (baseline) serum creatinine determination was recorded in the Department of Urology at the time of the initial oncological diagnosis, while the second serum creatinine sample was obtained in the Oncology Hospital after at least 90 days. If the eGFR level at the second determination was >60 mL/min/1.73 m^2^, the patients were excluded from the evaluation.

During the 2-year study period, the serum creatinine levels were repeatedly determined, in order to estimate eGFR (at least 4 determinations were performed every year; in most cases, the determinations were performed monthly or every 2 months). We did not use albuminuria and proteinuria as markers of CKD, due to the potential interference of toxic renal effects caused by some of the oncological therapies (please see Section 4).

We separately analyzed the patients with the three most frequent urological cancers (prostate, bladder and kidney), including the main demographic features (number, age at initial cancer diagnosis), nephrological features (eGFR at initial diagnosis and at final evaluation, presence of obstructive uropathy at initial diagnosis), oncological features, the most frequent co-morbidities and, most importantly, the 2-year mortality.

The assessment of the effect of cancer therapy on the kidney function was beyond the scope of our current analysis, as it was very difficult to perform subgroup analyses on relatively small samples of patients.

### Statistical Analysis

The continuous variables were expressed as the mean ± standard deviation (SD), while categorical variables were expressed as percentages. Univariate comparisons of baseline characteristics, using Student’s *t*-test or Chi-square test as appropriate, were performed for each group. All statistical tests were 2-tailed and a *p*-value < 0.05 was considered statistically significant. Odds ratios were calculated according to the methodology described by Altman in 1991, using a confidence interval of 95% [13]. We also performed the Mantel–Haenszel test to eliminate the confounding effect of the most significant co-morbidities (cardiovascular disease and diabetes mellitus) [14]. All the statistical analyses were performed using NCSS 2023 software (NCSS LLC, Kaysville, UT, USA).

## 3. Results

### 3.1. Impact of CKD on the Overall Mortality of the Patients

From the total group of 5831 patients, according to the selection criteria, we included 249 patients with the three most frequently encountered urological malignancies (prostate, bladder and renal cancer), with complete data available both from the Urology and Oncology Department. The prevalence of CKD in the patients with urological malignancies was 27.7%. The main demographic characteristics of the 249 patients are presented in Table 1.

We evaluated the impact of CKD on the overall mortality (all-cause mortality) of the patients from the entire group, and we found out that the odds ratio for the 2-year overall mortality of the three most frequent types of urological cancer patients with CKD compared with those without initial CKD was 1.75, with a 95% confidence interval between 1.03 and 2.97 (*p* < 0.05). After the elimination of the confounding effect of the most significant co-morbidities, the odds ratio for the 2-year overall mortality was 1.27 (95% confidence interval: 0.53–2.59; *p* < 0.05).

### 3.2. The Analysis of the Patients with Prostate Cancer

Next, we evaluated the data from the group of 146 patients with prostate cancer, including in our analysis the Prostate-Specific Antigen (PSA) values at the moment of the initial diagnosis, the distribution by tumor stage (localized, locally advanced or metastatic prostate cancer), the different forms of initial cancer therapy (radical prostatectomy, radiation therapy, hormone therapy or chemotherapy) and the most significant co-morbidities. The prevalence of CKD in this group was 16.4% (Table 2).

We can observe that a larger number of patients with obstructive uropathy at the moment of the initial cancer diagnosis were in the CKD group, mainly due to their more advanced cancer stage (62.5% of the CKD patients were in locally advanced and metastatic stages, compared with 52.5% in the control group, *p* < 0.05); many of the patients in advanced stages had acute/chronic urinary retention, and/or ureterohydronephrosis due to the invasion of the ureteral orifices. The average PSA value was significantly higher in the CKD group (213.7 ng/dL vs. 94.3 ng/dL in the control group), in concordance with the findings about the more advanced cancer stages.

Regarding the therapy type, a smaller proportion of the patients with metastatic disease from the CKD group benefited from the combined regimen of chemotherapy and androgen deprivation therapy (12.5% in the CKD group, compared with 19.7% in the control group), as they had low eGFR levels, which did not improve significantly after the resolving of the obstructive uropathy.

Finally, the 2-year overall mortality was significantly higher in the CKD group (62.5% vs. 39.3% in the control group); even after considering the effect of the more advanced stage, CKD remained the most significant predictor for higher mortality in patients with prostate cancer. The odds ratio for 2-year overall mortality of prostate cancer patients with CKD compared with those without initial CKD was 1.62, with a 95% confidence interval between 0.71 and 3.67 (*p* < 0.05). After the elimination of the confounding effect of the most significant co-morbidities, the odds ratio for the 2-year overall mortality was 1.21 (95% confidence interval: 0.28–5.23; *p* < 0.05).

Using the data from the CKD group, we further performed an analysis of the impact of the CKD stage (creating a group for patients with eGFR < 45 mL/min/1.73 m^2^, corresponding to stages G3b-G5, and a group for patients with eGFR ≥ 45 mL/min/1.73 m^2^, corresponding to stages G1-G3a). The odds ratio for the 2-year overall mortality was 3.57 in favor of the group with advanced CKD (stages G3b-G5), with a 95% confidence interval between 1.23 and 10.34 (*p* < 0.05).

### 3.3. The Analysis of the Patients with Bladder Cancer

In the following step, we evaluated the data from the group of 62 patients with bladder cancer, including the distribution by tumor stage, the different forms of cancer therapy and the co-morbidities. The prevalence of CKD was 32.3% (Table 3).

There were significant differences regarding the age at the moment of the oncological diagnosis: patients with CKD were significantly older than those without CKD (67.45 years vs. 62.17 years). The proportion of patients with obstructive uropathy was also higher in the CKD group (45% vs. 28.6%).

Tumor stage distribution was approximately the same in the two groups, with no significant differences between the percentages of patients with non-muscle-invasive, muscle-invasive or metastatic bladder cancer. A significantly lower proportion of the patients with CKD benefited from the systemic chemotherapy, as expected, explaining in part the higher mortality.

The 2-year mortality rate was almost double in the CKD group (80% vs. 45.2% in the control group). The odds ratio for 2-year mortality of bladder cancer patients with CKD compared with those without initial CKD was 2.74, with a 95% confidence interval between 0.83 and 8.98 (*p* < 0.05). After the elimination of the confounding effect of the most significant co-morbidities, the odds ratio for the 2-year overall mortality was reduced to 1.13 (95% confidence interval: 0.21–7.05, *p* = 0.53). 

### 3.4. The Analysis of the Patients with Renal Cell Carcinoma

Finally, we evaluated the data from the group of 41 patients with renal cancer, including the distribution by tumor stage, the distribution by oncological therapy and the co-morbidities. The prevalence of CKD in this group was 60.9%, the highest of the three types of urological cancers (Table 4).

We found significant differences regarding the mean age of the patients at the moment of initial cancer diagnosis between the two groups (68.52 years vs. 58.94 years). On the other hand, the proportion of patients with obstructive uropathy was similar in the two groups (12% vs. 12.5%), mostly due to bulky retroperitoneal masses and benign prostatic hyperplasia. However, the number of affected patients was significantly lower than the other two studied tumor locations, as expected.

A significantly higher proportion of the patients with CKD underwent nephron-sparing surgery, as expected (48% vs. 18.8%); moreover, this could be partly explained with the significantly higher proportion of patients diagnosed with localized renal cancer in the CKD group (64% vs. 43.8%). Despite this finding, the mortality rate was higher in the CKD group, confirming that CKD is a more important outcome predictor than cancer stage. The odds ratio for 2-year mortality of renal cell carcinoma patients with CKD compared with those without initial CKD was 1.25, with a 95% confidence interval between 0.41 and 3.84 (*p* < 0.05) (Table 5). After the elimination of the confounding effect of the most significant co-morbidities, the odds ratio for the 2-year overall mortality was reduced to 1.03 (95% confidence interval: 0.13–8.08, *p* = 0.67). 

## 4. Discussion

Most published clinical studies have analyzed the incidence of different forms of cancer in patient populations with chronic kidney disease [15,16,17,18,19,20]. However, just a few studies have addressed the subject the other way around, trying to evaluate the impact of chronic kidney disease on the overall survival and other patient outcomes [21,22,23].

In one of these investigations, Chinnadurai et al. conducted a matched-cohort study of CKD patients enrolled in the Salford Kidney Study, and found out that the kidney, bladder and prostate cancer had the highest prevalence; moreover, multivariate Cox regression has established a strong association between genito-urinary cancer and all-cause mortality [24].

In another study, published by Guo et al., it was demonstrated that CKD was significantly associated with different types of genito-urinary cancer, even after adjusting for confounding factors as different demographic variables, diabetes mellitus, hypertension, smoking status or BMI; additionally, CKD was established as an independent risk factor for overall survival [25].

The research team of Kitchlu et al. evaluated all the residents from the province of Ontario in Canada with data available on their eGFR and found out that the risk of kidney and bladder cancer was significantly increased in patients with CKD; cancer-related mortality and overall mortality were significantly higher in patients with CKD and urological cancers, compared to other locations of cancer [26].

In an article published by Tollefson et al., using a database of 10,099 patients who underwent radical prostatectomy for prostate cancer at the Mayo Clinic, with a median follow up of 10.2 years, it was shown that the eGFR value had no impact on biochemical recurrence or cancer progression, but it was a significant predictor for all-cause mortality, even after controlling for age, BMI, PSA doubling time, Gleason score and clinical stage [27].

Another large-scale study published by Ning et al. evaluated data of 136,790 patients undergoing radical prostatectomy from the US Nationwide Inpatient Sample, with a follow up of 10 years [28]. The study found out that patients with CKD had a significantly higher risk of postoperative complications and acute kidney injury, raising the need for careful evaluation of the kidney function before choosing a surgical intervention.

In the retrospective cohort study performed in 2006 by Huang et al. on 662 patients undergoing partial or radical nephrectomy for renal cell carcinoma, 26% of the patients had pre-existing CKD; after the surgery, the percentage of patients with CKD increased to 39% [29]. In addition to the type of surgical intervention (tumorectomy/partial nephrectomy vs. radical nephrectomy), other surgical techniques may play a role, including the total ischemia time, tumor size or location [9]. Even if the move to nephron-sparing surgery diminished the likelihood of a worsening of CKD, we still should consider the increasing prevalence of RCC and the older age of the population diagnosed as factors raising the need for systematic evaluation from the nephrologists [30].

A study published by Kim et al. analyzed 1855 patients who underwent radical nephrectomy for RCC between 1999 and 2011 in various centers in South Korea [31]. A multivariate regression analysis showed that preoperative CKD is an independent predictor for cancer-specific survival and overall survival; moreover, preoperative CKD was associated with more aggressive cancer features, requiring a more careful and frequent follow up.

Finally, the review article published by Saly et al. pointed out that risk factors such as hypertension, diabetes mellitus, obesity and smoking may contribute to the increased incidence of RCC in CKD patients, creating a specific oncogenic environment within the kidney [10]. Several pathways have been considered as playing a role in the oncogenesis process, including DNA damage and repair pathways, carcinogenic uremic toxins, chronic inflammation, oxidative stress and immunosuppression.

The impact of CKD on the mortality of patients with urological malignancies is variable, depending on several factors, like the severity of CKD, the presence of co-morbidities and general health status, which may play a significant role in determining outcomes of the patients [4,7].

The high mortality rates for all three urological malignancies observed in our study could be explained with the high incidence of CKD, the relatively high proportion of patients diagnosed in advanced stages (62.5% for prostate cancer, 45% for bladder cancer, 36% for renal cancer), the presence of significant concomitant cardiovascular disease and the specific situations encountered during the COVID-19 pandemic, when the follow up and therapy of some patients were significantly delayed. A future research paper from our collective will address this issue of the impact of COVID-19 on the diagnosis and therapy of patients with urological malignancies.

Although proteinuria has a well-defined prognostic role for the evaluation of renal function after renal surgery for renal cell carcinoma, and despite the fact that the nephrology guidelines recommend the dosing of albuminuria as a marker for CKD, we know that in many patients, including those with diabetes mellitus, eGFR may decrease before albuminuria [32]. On the other hand, the damage induced with the toxic effect of some oncological therapies is more frequently associated with acute kidney injury, because of acute tubule–interstitial nephritis [33]. Considering the level of bias potentially induced with these factors, we decided to not use albuminuria in our evaluation of CKD.

The contribution of obstructive uropathy to the overall mortality was difficult to assess, especially from a clinical perspective: even though many of the patients had documented obstruction of the upper urinary tract only for a limited period of time, we should consider that interstitial fibrosis and tubular atrophy are generally initiated after 2 weeks of continuous obstruction, but they do not disappear after the release of the obstruction [34].

Our study had several limitations, including a relatively low sample size for the groups of patients with bladder cancer and renal cancer, and its retrospective character. Moreover, the heterogeneity of our cohort could have an impact on the final outcomes, as we did not analyze variables such as the age, gender or body mass index (BMI), among others. Finally, during the study, we did not look for possible correlations between the progression of CKD and the oncological outcomes.

## 5. Conclusions

CKD is a significant predictor of the outcome and overall mortality in urological cancer and should be considered when recommending the most appropriate type of oncological therapy.

The development of newer, more effective cancer therapies has increased the proportion of patients with longer survival, but unfortunately, many of these therapies may be nephrotoxic.

A close collaboration between urologists, oncologists, radiation therapy specialists and nephrologists can help optimize the treatment decisions and management strategies. Moreover, the prevention, early detection, long-term follow up and specific therapy of CKD in this category of patients require the skills of an experienced nephrologist as part of the oncological multidisciplinary team [33].

## Figures and Tables

**Table 1 jpm-13-01572-t001:** The demographic characteristics of the two groups.

	CKD Group	Control Group	*p*
N	69	180	<0.05
Age in years at initial cancer diagnosis (±SD)	67.13 (±9.49)	62.7 (±9.95)	<0.05
Male/female ratio	4.75	8	<0.05
Two-year mortality	46%	75%	<0.05

**Table 2 jpm-13-01572-t002:** The main clinical parameters of the patients with prostate cancer.

	CKD Group	Control Group (No CKD at Diagnosis)	*p* (Statistical Significance)
Main demographic, nephrological and oncological characteristics			
N	24	122	
Age in years at initial cancer diagnosis (±SD)	65.42 (±9.07)	63.05 (±9.68)	NS
eGFR at diagnosis (±SD)	24.51 (±4.48)	89.21 (±17.76)	<0.05
eGFR at final evaluation (±SD)	43.38 (±13.03)	89.95 (±13.98)	<0.05
Obstructive uropathy at cancer diagnosis (%)	6 (25%)	18 (14.7%)	<0.05
PSA at diagnosis (ng/dL) (±SD)	213.7	94.3	<0.05
Two-year mortality (%)	15 (62.5%)	48 (39.3%)	<0.05
Distribution by tumor stage			
Localized (%)	9 (37.5%)	58 (47.5%)	NS
Locally advanced (%)	6 (25%)	31 (25.5%)	NS
Metastatic (%)	9 (37.5%)	33 (27.0%)	NS
Distribution by oncological therapy			
Radical prostatectomy (%)	3 (12.5%)	20(16.4%)	NS
Radiotherapy (%)	6 (25%)	38 (31.1%)	NS
Hormone therapy (%)	12 (50%)	40 (32.8%)	<0.05
Chemotherapy + Hormone therapy (%)	3 (12.5%)	24 (19.7%)	<0.05
Co-morbidities			
Cardiovascular disease (%)	11 (45.8%)	55 (45.1%)	NS
Diabetes mellitus (%)	2 (8.3%)	13 (10.7%)	NS
Other (%)	4 (16.7%)	16 (13.1%)	NS

NS = Not statistically significant.

**Table 3 jpm-13-01572-t003:** The main clinical parameters of the patients with bladder cancer.

	CKD Group	Control Group (No CKD at Diagnosis)	*p* (Statistical Significance)
Demographic, nephrological and oncological characteristics			
N	20	42	
Age at initial cancer diagnosis (years) (±SD)	67.45 (±9.31)	62.17 (±9.24)	<0.05
eGFR at diagnosis (±SD)	45.4 (±15.83)	88.8 (±12.62)	<0.05
eGFR at final evaluation (±SD)	39.01 (±15.00)	85.25 (±15.02)	<0.05
Obstructive uropathy at cancer diagnosis (%)	9 (45%)	12 (28.6%)	<0.05
Two-year mortality (%)	16 (80%)	19 (45.2%)	<0.05
Distribution by tumor stage			
Non-muscle-invasive (%)	11 (55%)	25 (59.5%)	NS
Muscle-invasive (%)	7 (35%)	13 (31%)	NS
Metastatic (%)	2 (10%)	4 (9.5%)	NS
Distribution by oncological therapy			
Transurethral resection (%)	11 (55%)	25 (59.5%)	NS
Bladder instillation therapy (%)	6 (30%)	13 (31%)	NS
Radical cystectomy (%)	3 (15%)	7 (16.7%)	NS
Radiotherapy (%)	4 (20%)	6 (14.3%)	NS
Chemotherapy (%)	2 (10%)	10 (23.8%)	<0.05
Co-morbidities			
Cardiovascular disease (%)	12 (60%)	23 (54.8%)	NS
Diabetes mellitus (%)	4 (20%)	6 (14.3%)	NS
Other (%)	4 (20%)	11 (26.2%)	NS

NS = Not statistically significant.

**Table 4 jpm-13-01572-t004:** The main clinical parameters of the patients with renal cell carcinoma.

	CKD Group	Control Group (No CKD at Diagnosis)	*p* (Statistical Significance)
Demographic, nephrological and oncological characteristics			
N	25	16	
Age at initial cancer diagnosis (years) (±SD)	68.52 (±9.78)	58.94 (±12.31)	<0.05
eGFR at diagnosis (±SD)	51.26 (±12.82)	76.5 (±19.95)	<0.05
eGFR at final evaluation (±SD)	46.2 (±11.28)	77.89 (±11.74)	<0.05
Obstructive uropathy at cancer diagnosis (%)	3 (12%)	2 (12.5%)	NS
Two-year mortality (%)	15 (60%)	8 (50%)	<0.05
Distribution by tumor stage			
Localized (%)	16 (64%)	7 (43.8%)	NS
Locally advanced (%)	4 (16%)	4 (25%)	NS
Metastatic (%)	5 (20%)	5 (31.2%)	NS
Distribution by oncological therapy			
Radical nephrectomy (%)	8 (32%)	8 (50%)	<0.05
Partial nephrectomy (%)	12 (48%)	3 (18.8%)	<0.05
Immunotherapy (%)	5 (20%)	5 (31.2%)	NS
Co-morbidities			
Cardiovascular disease (%)	5 (20%)	10 (62.5%)	<0.05
Diabetes mellitus (%)	2 (8%)	3 (18.8%)	NS
Other (%)	2 (8%)	2 (12.5%)	NS

NS = Not statistically significant.

**Table 5 jpm-13-01572-t005:** The odds ratio for the 2-year overall mortality of the three types of tumors studied, including the adjusted values after eliminating the confounding effect of cardiovascular disease and diabetes mellitus.

	Odds Ratio	95% Confidence Interval	Adjusted Odds Ratio	Adjusted 95% Confidence Interval
Prostate cancer	1.62	0.71–3.67	1.21	0.28–5.22
Bladder cancer	2.74	0.83–8.98	1.13	0.21–7.05
Renal cell carcinoma	1.25	0.41–3.84	1.03	0.13–8.08

## Data Availability

Data available on request.

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
