# Peer review of "The Impact of Chronic Kidney Disease on the Mortality Rates of Patients with Urological Cancers—An Analysis of a Uro-Oncology Database from Eastern Europe"

_jpm, 2023, doi:10.3390/jpm13111572_

Round 1

Reviewer 1 Report

Comments and Suggestions for Authors

I have read the paper with great interest. The observational retrospective cohort study aims to evaluate the effect of CKD on overall survival in patients who suffer from genitourinary cancer including prostate, bladder, and renal cancers.

The study contains 146 prostate, 62 bladder, and 41 renal cell carcinoma patients.

The authors declare that CKD and non-CKD groups' tumor stages and comorbidities are comparable.

As a conclusion, the authors state that CKD significantly predicts outcome and overall mortality in urological cancer.

My thoughts are as follows.

1-The heterogeneity of the cohort is a major problem in having standard outcomes. Multiple variables including gender, BMI, and stage of CKD etc.. may affect the final outcomes.

2- CKD is accepted as a major risk factor for coronary artery disease and related mortality. Factors such as the duration of CKD and being on a hemodialysis program may have an impact on mortality.

3- I am curious about whether gender and stage of CKD have an impact on overall survival. What do the authors think about this issue?

Sincerely yours,

Author Response

Dear reviewer,

Thank you very much for your appreciations and observations. I will try to answer to all your questions and comments, as follows: 

1-The heterogeneity of the cohort is a major problem in having standard outcomes. Multiple variables including gender, BMI, and stage of CKD etc.. may affect the final outcomes.

Answer: Your observation is correct, the cohort heterogeneity was an important issue, with impact on our results. Despite this fact, we think that our research adds some new value to the already published data, providing insight on the outcomes of a real-life population sample. We have included a new paragraph addressing the issue of cohort heterogeneity in the "Discussion section".

2- CKD is accepted as a major risk factor for coronary artery disease and related mortality. Factors such as the duration of CKD and being on a hemodialysis program may have an impact on mortality.

Answer: We completely agree that CKD is a major risk factor for coronary artery disease and for subsequent mortality. Regarding our study population, please note that the patients were diagnosed with CKD at the entrance of the 2-year observation period, and that the patients with previously diagnosed CKD (some of them even on hemodialysis) were excluded from the selection. In order to clarify this aspect, we have updated the paragraph on patient selection from the "Material and Methods" section. 

3- I am curious about whether gender and stage of CKD have an impact on overall survival. What do the authors think about this issue?

Answer: Following your suggestion, we have performed separate evaluations, considering the gender and the stage of CKD. We have obtained the following results:

- For the gender comparison we have analyzed only the groups with bladder cancer and renal cell carcinoma, as the prostate cancer group had only men, while the total group was biased by the large proportion of patients with prostate cancer. The odds ratio for 2-year overall mortality had a value of 3.5 in favor of men in the renal cell carcinoma group, and a value of 2.2 in the bladder cancer group, but the statistical significance was poor (0.15 and 0.48, respectively).

- For the stage of CKD (G1-G3a vs. G4-G5), we have calculated an odds ratio of 3.57 in favor of G4-G5 (P = 0.0189) for the total group, of 2.29 for the prostate cancer group (P = 0.3455), of 2.25 (P = 0.3305) for the renal cell carcinoma group, and of 2.32 (P = 0.1387) for the bladder cancer group.

- Considering the low statistical significance of most of the above results, we have decided to include in the manuscript only the results for the CKD stage of the total group.

Once again, many thanks for your review!

Reviewer 2 Report

Comments and Suggestions for Authors

Major revision:

 I read with great interest the manuscript on The Impact of Chronic
Kidney Disease on The Mortality Rates of Patients with Urological Cancers.

 Below my consideration:

- Tables should be revised, a NS result it is not always stated when present.
(authors should decide if they want to use NS or *.)
- The p-value of the multivariate analysis is missing.
- Is the two year mortality an overall mortality or a Cancer specific
mortality? please clarify.
- Line 276-278 authors may rely on the following paper: doi: 10.23736/S2724-6051.21.04308-1

Author Response

Dear reviewer,

Thank you very much for your comments and observations. You will find below our punctual answers:

- Tables should be revised, a NS result it is not always stated when present.
(authors should decide if they want to use NS or *.)

Answer: We have revised the tables and included NS, instead of *, where it was appropriate.

- The p-value of the multivariate analysis is missing.

Answer: We have included the p values of the multivariate analysis.

Is the two year mortality an overall mortality or a Cancer specific
mortality? please clarify.

Answer: In all instances, we have presented data about overall mortality.

- Line 276-278 authors may rely on the following paper: doi: 10.23736/S2724-6051.21.04308-1

Answer: Thank you for your suggestion, we have included the paper in our references.

Once again, many thanks for your review!

Round 2

Reviewer 2 Report

Comments and Suggestions for Authors

The authors have addressed my major concerns. i suggest pubblication